# Development of Therapeutic Vaccine for Chronic Hepatitis B: Concept, Cellular and Molecular Events, Design, Limitation, and Future Projection

**DOI:** 10.3390/vaccines10101644

**Published:** 2022-09-30

**Authors:** Sheikh Mohammad Fazle Akbar, Mamun Al Mahtab, Sakirul Khan, Osamu Yoshida, Yoichi Hiasa

**Affiliations:** 1Department of Gastroenterology and Metabology, Graduate School of Medicine, Ehime University, Ehime 791-0295, Japan; 2Interventional Hepatology Division, Department of Hepatology, Bangabandhu Sheikh Mujib Medical University (BSMMU), Dhaka 1000, Bangladesh; 3Department of Microbiology, Oita University, Oita 879-5593, Japan

**Keywords:** chronic hepatitis B, vaccine therapy, molecular mechanisms, scope and limitation

## Abstract

Four decades have passed since the first usage of the therapeutic vaccine in patients with chronic hepatitis B (CHB). However, there is no approved regimen of vaccine therapy for the treatment of CHB. This is mainly attributable to faulty conception, an improper understanding of the cellular and molecular mechanisms of CHB, and the impaired design of vaccine therapy for CHB. With the advent of new techniques and a better understanding of cellular and molecular mechanisms underlying the genesis of CHB, the limitations and failures of previous regimens of therapeutic vaccines have been primarily understood. Additionally, the importance of immune therapy for treating millions of CHB patients and achieving the target of “Elimination of Hepatitis by 2030” has been focused on in the international arena. This has been amplified by the apparent limitation of commercially available antiviral drugs that are infinite in duration, endowed with safety concerns, and unable to cure liver damage due to their minimal immune modulation capacities. The proposed review article comprehensively discusses each of these points and proposes evidence-based approaches for viable types of vaccine therapy for the treatment of CHB.

## 1. Introduction

Hepatitis B virus (HBV) infection is global. Considerable insights into this virus have developed from the Bronze Age to the first quarter of the 21st century, especially after the discovery of HBV in the mid-1960s. Most HBV infections occur during birth or the perinatal period, or at pre-school age, and considerable numbers of these subjects are chronically infected by HBV (expressing hepatitis B surface antigen (HBsAg) and/or HBV DNA in the blood). The epidemiology of HBV has exposed that about 2 billion HBV-infected subjects reside in the world. They usually express antibodies to hepatitis B core antigen (HBcAg) (anti-Anti-HBc). However, about 1.7 billion of these 2 billion people control HBV infection, and they do not express HBV DNA or hepatitis surface antigen (HBsAg). They usually do not progress to cirrhosis of the liver (LC) or hepatocellular carcinoma (HCC). In fact, they usually do not transmit HBV infection to healthy persons [1]. Out of these 2 billion people, about 296 million express HBsAg and HBV DNA in their blood [2]. These 296 million estimated chronic HBV-infected subjects represent living and permanent reservoirs of HBV and transmit HBV to healthy persons; thus, the transmission cycle of HBV is maintained as human beings are possibly the only hosts of HBV. Among these 296 million people, about 12–25% (estimated 34 million to 70 million) of patients express evidence of liver damage (shown by elevated levels of alanine aminotransaminase (ALT)) in addition to being positive for HBsAg and HBV DNA [3,4]. In fact, most patients develop CHB during adolescence or after becoming adults, although they are infected with HBV much earlier. Considerable numbers of CHB patients expressing HBsAg, HBV DNA, and elevated ALT (an estimated 34–70 million) will eventually develop features of progressive hepatic fibrosis and its complication such as LC. LC is characterized by distortion of the lobular structure of the liver and alteration of hemodynamics. Patients with LC are prone to developing decompensated liver diseases with hepatic failure and HCC after one or two decades [5,6]. The impact of HBV infection is so devastating that an estimated 882,000 people died due to HBV-related liver diseases in 2019 (estimates of the World Health Organization (WHO)) (Figure 1A) [2].

Thus, it is apparent that if CHB patients can be properly treated, incidences of LC and HCC can be contained to a great extent, and the transmission of infection, the suffering of millions of people, and the public health burden of society can be reduced. Thus, the treatment and management of millions of CHB patients have become an urgent public health concern for the international community. Due to this massive problem with HBV infection, the global health sector strategy on viral hepatitis initiated the concept of “Elimination of Hepatitis by 2030” in 2016 [7]. The WHO has also set several targets for attaining this goal [8,9]. The treatment coverage should be raised to 80% by 2030 [8,9]. Thus, the WHO target indicates that several million CHB patients should be treated by 2030 (Figure 1B).

Several antiviral drugs have been developed during the last four decades to treat CHB patients, mainly including interferons (IFNs), their pegylated forms, and a group of drugs known as nucleoside analogs (NUCs). Both groups of drugs are endowed with antiviral properties, and some of these drugs have reported some beneficial effects in the containment of hepatic fibrosis in CHB patients [10,11,12]. However, due to the inherent limitations of these drugs, neither IFNs nor NUCs represent a drug of choice for the treatment of CHB; in fact, these drugs have not been able to stand the test of time [13,14,15,16] (Table 1).

From this perspective, there remains a pressing and urgent need to develop a proper regimen of new and novel therapeutics for CHB patients. Based on the genesis of cellular and molecular mechanisms underlying the development of CHB, it appears that immune therapy might be an evidence-based therapy for CHB patients. The first immune therapy trial was initiated by Dienstag et al. in CHB patients [17]. Pol et al. first reported the clinical utility of the HB vaccine for treating CHB patients in 1994, and the regimen was called “vaccine therapy” [18]. During the last 28 years, there have been several modifications of vaccine therapy for CHB regarding the nature, design, and vaccines. However, a proper regimen of vaccine therapy is yet to emerge that would be evidence-based, safe, of a finite duration, and efficacious to contain both HBV replication and liver damage.

In this review, the concept of vaccine therapy and the cellular and molecular basis of vaccine therapy are re-analyzed. Additionally, evidence is provided regarding the apparent failure of ongoing regimens of vaccine therapy for CHB. Finally, future projections about vaccine therapy are summarized with an objective to rationalize this innovative therapy for millions of CHB patients and attain the WHO goal of “Elimination of Hepatitis by 2030”.

## 2. Limitation of Commercially Available Antiviral Drugs

Two forms of antiviral drugs have been recommended for treating CHB patients: IFNs, which have been used since the mid-1980s, and NUCs, which have been available since the late 1990s. IFNs are not a drug of choice for the therapy of CHB patients due to significant concerns about their safety, parenteral l mode of administration, and cost [19,20]. Even without such limitations, IFNs could only contain HBV replication and liver damage in a minor percentage of CHB patients [21]. On the other hand, NUCs are endowed with several positive points for usage by CHB patients. These include their oral administration, which is comparatively cheaper than the administration of IFNs, and their safety, being endowed with high potency to induce HBV DNA reduction and negativity [22,23,24]. Even after inducing considerable optimism, NUCs have not been able to become a model drug because they should be used for an infinite duration, and this may be several years to life. Cessation of NUCs induces a rebound of HBV replication, and there may be a hepatic flare in some patients, which may bring fatal outcomes. NUCs are not effective in eliminating cccDNA. Thus, stoppage of NUCs induces replication and other complications, the immune modulatory capacity of NUCs is insignificant, and the role of NUCs in the progression of hepatitis and the genesis of fibrosis is limited [25,26,27] (Table 1).

## 3. Why Immune Therapy Became an Option for Treatment of CHB

The concept of immune therapy for treating CHB patients was optimized based on investigational information and evidence. HBV is a non-cytopathic virus. HBV DNA or its antigens cannot induce damage to the liver, such as inflammation, fibrosis, and carcinogenesis, although some investigators have claimed the role of hepatitis B X antigen in hepatocarcinogenesis by retrieving data from animal studies [28,29,30]. Thus, the mechanism of how pathological lesions are caused in CHB patients due to the impact of non-cytopathic HBV becomes a natural query. Investigators have shown that the host immunity following HBV infection is primarily responsible for HBV-induced liver damage. In this connection, the activity of different immunocytes was analyzed in animal models of HBV and CHB patients. Studies in animal models of HBV, such as HBV transgenic mice (HBV TM), revealed that HBV-antigen-specific immunity by HBV-related antigens was related to containment of HBV replication, and that the seronegativity of hepatitis B e antigen (HBeAg) and seroconversion of the antibody to HBeAg (anti-HBe) [29,30,31,32,33,34] are two significant events related to containment of HBV replication and liver damage. Circumstantial evidence also supported this concept in patients with CHB. Investigators checked this regarding the immunopathogenesis of CHB in two groups of CHB patients: one group with a high viral load and increased liver damage, and another group controlling HBV replication with contained liver damage. It was found that non-antigen-specific immunity may be detrimental for CHB patients, whereas antigen-specific immune induction in CHB patients resulted in viral containment and protection from liver damage [35,36,37].

## 4. The Advent of Vaccine Therapy for Treating CHB Patients

Once it was found that modulation of host immunity may have therapeutic potential in CHB patients, several investigators tried to upregulate host immunity using polyclonal immune modulators, such as interleukin (IL) 2, IL12, granulocyte-macrophage colony-stimulating factor, levamisole, and others. Most of the studies with polyclonal modulators were accomplished with commercially available drugs. Different protocols were used in CHB patients. Some of the studies reported initial optimism, but follow-up data were not provided. It was finally revealed that the polyclonal immune modulators were not safe, and that the efficacy was questionable [38,39,40,41,42].

By this time, the attention on the therapy of CHB patients was diverted towards the usage of HBV antigen-specific immune therapy in CHB patients. As a part of this concept, Pol et al. first used the HBsAg-based prophylactic vaccine to treat CHB patients in 1994 [18]. This study revealed that the HbsAg-based vaccine reduced HBV DNA and induced seroconversion to anti-HBe in some patients. As a prophylactic vaccine was used in this study, the therapy received its name, “vaccine therapy”.

## 5. Different Modes of Vaccine Therapies for CHB Patients (from 1994 to 2022)

### 5.1. HBsAg-Based Vaccine Therapy

There are some fundamental reasons why HbsAg-based vaccine therapy took the stage of antigen-specific immune therapy for CHB. HbsAg-based vaccines have been used in normal healthy subjects since the 1980s as prophylactic vaccines [43,44,45]. They are some of the safest vaccines in the global context. Additionally, all chronic CHB patients harbor HBsAg for more than six months, and the development of antibodies to HBsAg (anti-HBs) in CHB patients is usually regarded as recovery from CHB status. It was assumed that an HBsAg-based vaccine would be able to accomplish the production of anti-HBs and the ultimate cure for this pathological condition (Table 2).

### 5.2. Different Forms of HBsAg-Based Vaccine Therapy for Treating CHB Patients and Their Limitations

Various forms of HBsAg-based vaccines have been administered in CHB patients, the fundamental component of which was HBsAg. HBsAg-based vaccines have been used as protein forms [46,47,48] or with anti-HBs as antigen/antibody complex vaccines [49,50,51]. DNA was constituted that expressed HBsAg and was administered in CHB patients [52]. Mancini-Bourgine et al. used an HBV DNA vaccine in 10 patients with CHB. Seroconversion to anti-HBe was recorded in two patients. However, follow-up data are not available [53]. Although DNA vaccination may be an optimistic approach for innovative therapy against CHB, Cova et al. [54] concluded that DNA-based therapy should be optimized by conducting more trials, as the outcome in animal models of chronic HBV-infected preclinical studies could not be translated in CHB patients.

HBsAg-pulsed dendritic cells (DCs) were assessed regarding their immune modulatory capacities. HBsAg-pulsed DCs were also administered in patients with CHB [55,56,57]. Administration of HBsAg-pulsed DCs revealed a transient effect on HBV DNA, but a sustained effect could not be recorded.

In the meantime, a hypothesis has been advancing that the HBsAg-based vaccine may be more efficient if the usage of antiviral drugs can reduce the levels of HBV DNA. Thus, HBsAg-based vaccine therapy has also been used as part of combination therapy with antiviral drugs [58,59,60]. A short account of the HBsAg-based vaccine as vaccine therapy is shown in Table 3 as a ready reference.

### 5.3. Other Forms of Vaccine Therapy in Addition to HBsAg-Based Vaccine

The limitation of HBsAg-based vaccine therapy became evident after several clinical trials in CHB patients [46,47,48,49,50,51,52,53,54,55,56,57,58,59,60]. This is due to the fact that there is a paucity of information about the mechanisms of action of HBsAg-based vaccine therapy for CHB patients. In addition, the HBsAg-based vaccine may be able to induce HBsAg-specific immunity in CHB patients; however, recovery from the CHB state is related to HBcAg-specific immunity in CHB patients [35]. This has led to the use of other HBV-antigen-specific immune modulators for vaccine therapy despite using a preventive vaccine of HBsAg. Some of these modulators used in CCHB patients are shown in Table 4.

### 5.4. A Bird’s Eye View of the “Vaccine Therapy” in CHB Patients

A comprehensive view of vaccine therapy in CHB patents has been shown in the above three sections of the article (Section 5.1, Section 5.2 and Section 5.3). In addition to these, have the scope of limitations of vaccine therapy including other forms of innovative therapy for CHB patients [63,64,65,66].

## 6. Limitations of “Vaccine Therapy” in CHB Patients

About 28 years (1994 to 2022) have passed since the initiation of the first vaccine therapy in CHB patients. However, there has been no consensus about the clinical usage of any therapeutic vaccine for CHB until now. Elucidation of these points is essential for the development of any viable treatment option for CHB. As of today, it is challenging to provide accurate insights into why any regimen of vaccine therapy could not be rationalized to the scientific community and the millions of CHB patients. Drug development includes a complex procedure involving the combination of “Concepts”, “Evidence”, “Preclinical studies with animal models”, “phase I/II/III clinical trials”, and data of short-, mid-, and long-term prognoses. Vaccine-based therapies did not follow these procedures, with some exceptions. Immune therapy with the HBsAg-based vaccine was accomplished mostly in pilot studies, observational studies, or small-scale phase studies. After providing initial data at the end of treatment (EOT), in most cases, there were no publications about long-term follow-up. The proper design of immune therapy should be first authenticated by a concept; this should be followed by animal and preclinical studies. Additionally, there remains extreme heterogeneity among different protocols of vaccine therapy that have been accomplished during the last 28 years. Some of the studies used only HBsAg, whereas others used different forms of HBsAg. The dose was also variable and fluctuated from 10 micrograms to 40 micrograms. The duration of therapy also varied from three to twelve administrations of the HBsAg-based vaccine. Moreover, studies about the mechanisms of HBsAg-based vaccines are scarce [18,46,47,48,49,50]. Taken together, during the 1990s, HBsAg-based vaccine therapy inspired considerable optimism as a new and novel therapeutic for CHB patients. However, it could not stand the test of time as an independent supportive mode of therapy for CHB patients due to the amicable limitations of drug development. An account of the major limitations of HBsAg-based vaccines is given below:HBsAg-based vaccine therapy is not an evidence-based immune therapy for CHB patients. Immunity to HBsAg, such as anti-HBs, is an essential factor for controlling circulating HBV. Thus, the HBsAg-based vaccine is critical in the context of preventing HBV infection. However, treatment of CHB patients is related to the control of intracellular HBV DNA, especially cccDNA. This cannot be accomplished by the anti-HB response initiated by the HBsAg-based vaccine.Almost all investigators have used their own protocol for accomplishing vaccine therapy. Thus, the dose and duration of immunization have varied among protocols. This makes it difficult to assess the real implication of the optimum dose or duration of HBsAg that should be used in vaccine therapy.Almost all studies about vaccine therapy in CHB patients have been accomplished as pilot studies or observational studies, or as clinical trials of a limited spectrum. Thus, phase I/II/III clinical trials along with a follow-up study of several years with vaccine therapy remain to be accomplished.In almost all cases of vaccine therapy for CHB, there has been no publication regarding the short- or long-term follow-up information of this therapy.In most cases of vaccine therapy in CHB patients, the mechanism of action has been elucidated. The query remains regarding the presence of HBsAg in all CHB patients. Further, what specific functions can be achieved by administering HBsAg? Thus, there is a need to show differences in the immune modulation of circulating HBsAg and administered HBsAg-based vaccines. This has not been properly dissected.

## 7. Role of HBcAg as a Component of Vaccine Therapy

A group of studies on both HBV TM and patients with CHB revealed that HBcAg-specific immunity is essential for the control of HBV replication and containment of liver damage. Recently, a drug that contains both HBsAg and HBcAg (NASVAC) was used via the nasal and parenteral routes in treatment-naïve CHB patients as well as NUC-treated CHB patients [67,68,69,70]. This group of studies systematically provided data for the end of treatment (EOT), 24 weeks after the EOT [64], and two and three years after the EOT [71,72] in treatment-naïve CHB patients after being treated with an HBsAg/HBcAg-based therapeutic vaccine. Negativity of HBV DNA in the sera and normalization of serum ALT were shown in more than two thirds of the patients in these studies. Additionally, these studies revealed that HBcAg-specific immunity was induced by the HBsAg/HBcAg-based vaccine in the treatment-naïve CHB patients. Although this vaccine provided optimistic data about vaccine therapy, multicenter studies should be conducted to validate these outcomes. Further, the use of HBsAg/HBcAg has been shown in patients treated with NUCs, and this approach has shown the possibility of a functional cure for CHB [73].

## 8. Positive Sides of Vaccine Therapy for the Treatment of CHB Patients

Despite several pertinent limitations of vaccine therapy, there is considerable optimism about this therapy in CHB patients. The experiences of the last 28 years about vaccine therapy have provided important insights into the pros and cons of vaccine therapy. The inspiring points of vaccine therapy and logic for optimism are summarized below:Almost all protocols of vaccine therapy against CHB have shown their excellent safety profiles.Most of the projects of vaccine therapy have been endowed with antiviral potentials in CHB patients.Some protocols of vaccine therapy have shown the capacity of therapy to induce HBeAg negativity or anti-HBeAg seroconversion.Normalization of ALT has been reported by most of the designs of vaccine therapy.

## 9. Future Projections of Vaccine Therapy for CHB Patients

If the scopes and limitations of vaccine therapy are critically analyzed, it appears that vaccine therapy is safe and endowed with antiviral and liver-protecting capacity. Most vaccine therapies have also indicated their finite nature and suppression of HBV DNA. Now, it is time to find out the means to optimize vaccine therapy to retrieve the maximum benefit in CHB patients. The following points are worth mentioning so that an acceptable regimen of vaccine therapy can be generated for CHB patients:Nature of the antigen: various pieces of evidence indicate that only HBsAg-based vaccine therapy may not be an optimum type of vaccine therapy as HBsAg-based immunity is unlikely to have an impact on intracellular HBV DNA and cccDNA.In spite of using HBsAg-based vaccine therapy, more attention should be directed to the use of HBcAg or other HBV-related antigens as candidate antigens for vaccine therapy.Studies about vaccine therapy should be properly designed regarding the dose and duration of therapy.Studies should be formulated to provide safety and efficacy information for the end of treatment (EOT) and after the EOT for a prolonged duration (data from over one to five years).When antiviral and liver protection is recorded by vaccine therapy, the underlying mechanisms should be explored.

## 10. Conclusions

We are standing at the crossroads of development of a prosperous drug for CHB. If CHB patients remain untreated, many of these patients will develop complications such as LC and HCC, which will impart a heavy burden on both the patients and the healthcare delivery system. It is worth mentioning that most CHB patients reside in developing and resource-constrained countries. Thus, we need drugs for CHB that will be safe, comparatively cheaper, and of a finite duration. Definitely, the drugs should contain the progression to LC and HCC. The current commercial drugs available in markets (IFNs and NUCs) could not properly attain their assigned goal of treating CHB patients for the containment of HBV DNA and control of elevated ALT, although these have shown benefits in some CHB patients. Regarding the role of NUCs in attaining functional care, the outlook is not so hopeful. In addition, IFNs and NUCs are repurposed drugs, and it seems unlikely that commercially available antiviral drugs will resolve the problem of CHB patients from a global perspective, especially for the “Elimination of Hepatitis by 2030” [68,69]. On one hand, HBV is a non-cytopathic virus, and liver diseases are caused by aberrant immunity. On the other hand, there is a need for finite therapy considering the prevalence of CHB in developing countries. From this perspective, vaccine therapy appears to be a potential evidence-based candidate for the treatment of CHB patients within a finite duration. The challenge is to retrieve maximum benefits by optimizing the nature of the vaccines and the duration of the therapy and releasing data of a follow-up study. Multicenter studies and international collaborations are essential for attaining these goals. Finally, a new mode of therapy with different combinations of antivirals and vaccine therapy may be beneficial for CHB patients.

## Figures and Tables

**Figure 1 vaccines-10-01644-f001:**
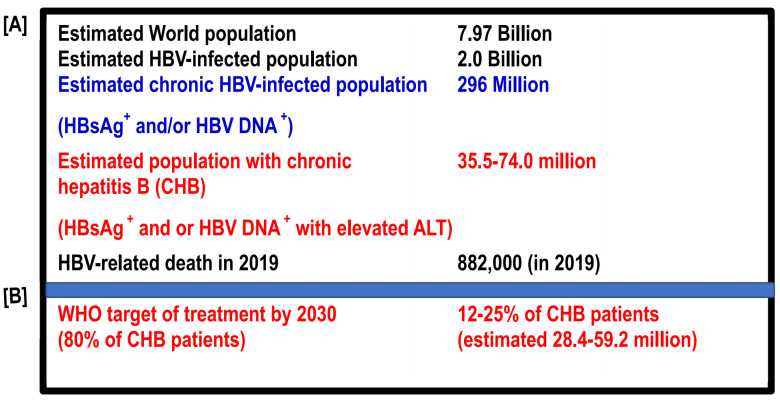
Spectrum of HBV infection in the world and targets of the World Health Organization to achieve “Elimination of Hepatitis by 2030” [8,9]. HBV, hepatitis B virus; CHB, chronic hepatitis B; HBsAg, hepatitis B surface antigen; ALT, alanine aminotransferase; WHO; World Health Organization.

**Table 1 vaccines-10-01644-t001:** Limitations of commercially available antiviral drugs and possible underlying causes.

Limitation of IFNs and NUCs	Possible Causes
IFNs and NUCs are unable to eradicate or substantially control cccDNA that acts as a template for HBV replication.	Interferons and nucleoside analogs are repurposed antiviral drugs developed for other viral infections. Thus, these drugs are not evidence-based drugs for CHB.
The liver-protecting capacities and potency to block progression to LC and HCC are nominal for IFNs and NUCs.	The immune modulatory capacity and the nature of immune modulation of IFNs and NUCs are not purpose-oriented.
NUCs is an infinite mode of therapy, and there is a need to continue NUC intake for years or even for life.	The half-life of NUCs is, at best, 60 h. Thus, the effect of NUCs would be of limited duration. Cessation of NUCs will favor a milieu of HBV replication.
NUCs are not patient-friendly for developing and resource-constrained countries.	The infinite usage of NUCs along with periodic check-ups is unfriendly to developing countries that harbor 80% of CHB patients.
Use of NUCs confuses patients in developing countries.	NUCs induce HBV DNA negativity after usage for a short duration. Many CHB patients consider this as a remedy from the disease and give up taking medication. However, the drug must be taken for a prolonged duration.

**Table 2 vaccines-10-01644-t002:** HBsAg-based vaccine as the ideal candidate for “vaccine therapy” of CHB patients.

Parameters	Explanations
Availability	The HBsAg-based vaccine has been used as a preventive vaccine for HBV infection since the 1980s and is available commercially around the world.
Safety	The safety of the HBsAg-based vaccine has been validated in millions of individuals.
Scientific rationality	Loss of HBsAg and development of antibodies to HBsAg (anti-HBs) have been regarded as a complete cure for HBV infection. It was assumed that the HBsAg-based vaccine will be able to accomplish immune modulation in favor of anti-HBs.

**Table 3 vaccines-10-01644-t003:** Different forms of HBsAg-based vaccine therapies.

Different Candidates of HBsAg-Based Vaccine Therapy	References
1. HBsAg-based protein vaccine	[46,47,48]
2. HBsAg/anti-HB-based antigen/antibody complex vaccine	[49,50,51]
3. HBsAg-based DNA vaccine	[53,54]
4. HBsAg-based cellular vaccine	[55,56,57]
5. HBsAg-based vaccine as part of combination therapy with antiviral drugs	[58,59,60]

**Table 4 vaccines-10-01644-t004:** Vaccine therapy for CHB patients with therapeutic vaccines other than HBsAg-based preventive vaccines.

Vaccine Therapy in CHB Patients Using Therapeutic Vaccines Other than HBsAg-Based Vaccine Therapy	References
1. Epitope-based vaccine	[61]
2. Lipoprotein-based vaccine	[62]

## Data Availability

Not applicable.

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
