# Peer review of "Development of Therapeutic Vaccine for Chronic Hepatitis B: Concept, Cellular and Molecular Events, Design, Limitation, and Future Projection"

_vaccines, 2022, doi:10.3390/vaccines10101644_

Round 1

Reviewer 1 Report

Akbar reviewed the development of a Therapeutic Vaccine for Chronic Hepatitis B explaining the concept of immune therapy and the different modes of vaccine therapies.
The manuscript’s strengths are based on the fact that this review illustrates concepts that lead to the understanding of vaccine therapy but it makes sense only if accompanied by other reviews that go into the details of the results obtained so far with vaccine therapy, otherwise,  the greatest weakness of the work is precisely the lack of these data.
Please correct some typo errors. 

Author Response

Response to Reviewer 1

Comments and Suggestions for Authors

Akbar reviewed the development of a Therapeutic Vaccine for Chronic Hepatitis B, explaining the concept of immune therapy and the different modes of vaccine therapies. The manuscript’s strengths are based on the fact that this review illustrates ideas that lead to the understanding of vaccine therapy, but it makes sense only if accompanied by other studies that go into the details of the results obtained so far with vaccine therapy; otherwise, the greatest weakness of the work is precisely the lack of these data. Please correct some typo errors. 

Response to the Reviewers:

Thank you very much for your constructive comments. As per the recommendation of the Reviewer, the reviews about “vaccine therapy” for chronic hepatitis B.

The new reviews are shown in the body of the manuscript (Page 6; subsection 5.4)

Additionally, the article has been submitted to the English Checking Authority of the journal. The revised manuscript has been checked for English usage and Layout adjustment (English Checking Certificate of MPDI: attached).

The new references have been shown by 63-66.

Reviewer 2 Report

The review by Akbar et al. aims to shed light in vaccine-based therapies against chronic hepatitis B through the years and give perspectives for future usage. Although the authors have made efforts in properly structuring the review, I am afraid that it suffers serious flaws listed below:

1.       The English language is unacceptable; the review must be proofread by a native speaker.

2.       In many parts, there is a considerable lack of precision and explanation

3.       It is hard for the reader to understand why vaccine-based therapies were not developed; what exactly did not work? Give data. What could make them usable in future? Justify!

4.       Many statements need to be supported by proofs (references)

5.       I felt a considerable negativism against NUCs. Although they cannot eliminate HBV, they work pretty well in controlling infection and liver disease progression.

Here are some precise comments (excluding the English grammar):

-          Lines 36-38: what is the difference between the 2 billion HBV-infected people and the 296 million expressing HBsAg and HBV DNA? Are these 2 billion people still carrying the virus or the great majority of them got infected at some point but have spontaneously cleared the virus? Of these 2 billion, approx. 300 million became chronic carriers. This also concerns figure 1.

-          Line 47: better distinguish liver cirrhosis than simple fibrosis!

-          Line 58: less than 1% of the deserving CHB patients were under some treatment. This is unbelievable especially without a precise citation/reference.  

-          Line 72-73: due to the inherent limitations of these drugs… explain! What are the limitations?

-          Line 91-92 the title of the paragraph is not clearly presented.

-          Line 96: parental is indeed PARENTERAL

-          Line 97: a minor percentage of CHB patients… What is the exact %? Also, in this respect you have not mentioned pegIFN

-          Table 1. The limitations are in my opinion exaggerated. Give references. Although repurposed, NUCs work well in lowering HBV DNA and improving patient’s condition. Similar to HIV, they should be taken infinitely but this blocks the progression of the disease.

-          Line 109: this whole paragraph should be rewritten. Line 111: what do you mean HBV is a non-cytopathic virus? HBV infection itself induces inflammation of the liver which plays role in the pathogenesis. Line 118: immunocytes?

-          Line 133: polyclonal immune modulators. Examples ? Develop more.

-          Line 138: considerable success. Which is…?

-          Lines 156-158: No sense… DNA approach should be better explained. The dendritic cells are pulsed with antigen, not the other way…

-          Line 166: explain the limitations

-          Lines 173-175: problem in texts. Why there has been no consensus? Explain!

-          Line 179: extreme heterogeneity… Explain!

-          Line 201-202: The fact that the mechanism of action has been elucidated is a limitation???

-          Line 210: optimistic data: give the information about that data! What were the results?

-          Line 252-254. I do not agree that NUCs do not stop progression to CHC. Proofs must be given.

Author Response

Response to Reviewer 2

Thank you very much for such an elaborative and critical review of the article. I am happy to mention that all the concerns raised by the Reviewer have been responded to. The responses have been given for each of the answers. 

Yellow Shading has shown the responses to the Reviewer and all sorts of modifications.

Comments and Suggestions for Authors

The review by Akbar et al. aims to shed light on vaccine-based therapies against chronic hepatitis B through the years and give perspectives for future usage. Although the authors have made efforts to adequately structure the review, I am afraid that it suffers serious flaws listed below:

  1. The English language is unacceptable; a native speaker must proofread the review.

Response: The English language has been checked by the professional English language service provided by the journal. 

  1. In many parts, there is a considerable lack of precision and explanation.

Response: These areas have been addressed in the revised manuscript

  1. It is hard for the reader to understand why vaccine-based therapies were not developed; what exactly did not work? Give data. What could make them usable in the future? Justify!

Response: Drug development includes a complex procedure of a combination of “Concepts,” “Evidence,” “Preclinical studies with the animal model,” “phase I/II/III clinical trial,” and providing data of short, middle, and long-term prognosis. Vaccine-based therapies did not follow these procedures, with some exceptions. Immune therapy with HBsAg-based vaccine was accomplished mainly as a pilot study, observational study, or small-scale phase study. After providing initial data at the end of treatment (EOT), in most cases, there was a publication about long-term follow-up. The proper design of immune therapy should be first authenticated by concept; this should be followed by animal and preclinical studies. Also, there remains heterogeneity among different protocols. Thus, an accepted protocol of immune therapy is yet to emerge in the clinic. This has been shown in the Text (6. Limitations of “vaccine therapy” in CHB patients).  

  1. Many statements need to be supported by proofs (references)

Response: References have been provided in different statements. 

  1. I felt considerable negativism against NUCs. Although they cannot eliminate HBV, they work pretty well in controlling infection and liver disease progression.

Here are some precise comments (excluding the English grammar):

Lines 36-38: what is the difference between the 2 billion HBV-infected people and the 296 million expressing HBsAg and HBV DNA? Are these 2 billion people still carrying the virus, or did the great majority of them get infected at some point but have spontaneously cleared the virus? Of these 2 billion, approx. Three hundred million became chronic carriers. This also concerns figure 1.

Response: According to the estimates of the World Health Organization, about 2 billion people in the world have been infected by HBV at some point in their life. Out of these vast numbers, about 1.7 billion control HBV infection, and they also do not transmit HBV to healthy individuals. About 296 million people harbor HBsAg and HBV and share the HBV with healthy persons. Currently, there is a management strategy for 1.7 billion HBV-infected people who do not express HBsAg and HBV DNA. Management and treatment are recommended for 296 million people who are chronic HBV-infected. A considerable number of these 296 million people also develop cirrhosis of liver and liver cancer. This has been described in detail in reference 1. Please check the data of WHO. Global Hepatitis Report 2017. Geneva: World Health Organization; 2017http://apps.who.int/iris/bitstream/10665/255016/1/9789241565455-eng.pdf

To make these data more amicable for the readers, this has been clarified in INTRODUCTION (Line 37-42)

Line 47: better distinguish liver cirrhosis from superficial fibrosis!

Response: This has been described in the revised Text (Line 52-55). 

Line 58: less than 1% of the deserving CHB patients were under some treatment. This is unbelievable, especially without a precise citation/reference.  

Response: I completely honor your conception, but this is the fact. Please check Table 1 given below; This is an official declaration of the WHO.

Linkage:

Line 72-73: due to the inherent limitations of these drugs, explain! What are the limits?

Response: The limitations of IFNs and NUCs have been described in detail in Table 1. I have been citing those again for ready references. The inherent limitations include the fact that IFNs and NUCs are unable to eradicate or substantially control cccDNA that acts as the template for HBV replication; the liver-protecting capacities and potency to block progression to LC and HCC are nominal for IFNs and NUCs; NUCs are an infinite mode of therapy, and there is need to continue NUC intake for years or even for life; Use of NUCs confuse patients of developing countries. Also, Table 1 has been cited at the place of this statement.

Line 91-92, the title of the paragraph is not clearly presented.

Response: This has been changed and simplified.

Line 96: parental is indeed PARENTERAL

Response: I am sorry for the mistake. Thank you very much for specifying this. This has been corrected in the revised manuscript.

Line 97: a minor percentage of CHB patients… What is the exact %? Also, in this respect, you have not mentioned begin.

Response: This is an essential issue regarding the use of IFNs in CHB patients. The reaction to IFN will depend on various factors, such as the evaluation of parameters, such as HBV DNA or quantification of HBsAg. Studies have found diverse data about this. Also, the point at which the evaluation was done is also essential to provide a percentage. I am citing an article that was published in August 2022 by Anna Lok. She documented data about the efficacy of IFN and showed that it varies from 0.38% to 9.66% to 13% to 33% based on the marker of HBV that was assessed and also the duration of the assessment. Several other studies also showed similar diverse data based on protocols. Linkage: https://www.uptodate.com/contents/pegylated-interferon-for-treatment-of-chronic-hepatitis-b-virus-infection.

Also, pre-treatment HBsAg was an important indicator of treatment success. If this was high, the response was recorded in 0% of patients. Wu et al. HBsAg quantification predict off-treatment response to interferon in chronic hepatitis B patients: a retrospective study of 250 cases. BMC Gastroenterology 2020;20;121

Table 1. The limitations are, in my opinion, exaggerated. Give references. Although repurposed, NUCs work well in lowering HBV DNA and improving the patient’s condition. Similar to HIV, they should be taken infinitely, but this blocks the progression of the disease.

Response: The limitations that have been shown in Table 1 may appear exacerbated. I completely agree with the Reviewer that if the drug is taken for life-like HIV, complications like HCC can be delayed but not contained. Five limitations of NUCs have been cited in Table 1. The efficacy of NUCs may be acceptable if life-long treatment can be ensured, like HIV, in which international organization helps to get the drug. But, for CHB, life-long usage of NICs on the basis of present realities seems impossible as most of the patients with CHB reside in developing countries, and they cannot comply with the life-long-treatment. 

Line 109: this whole paragraph should be rewritten. Line 111: what do you mean HBV is a non-cytopathic virus? HBV infection itself induces liver inflammation, which plays a role in the pathogenesis. Line 118: immunocytes?

Line 133: polyclonal immune modulators. Examples? Develop more.

Response: This has been described in more detail. 

Line 138: considerable success. Which is…?

Response: Described in more details

Lines 156-158: No sense… DNA approach should be better explained. The dendritic cells are pulsed with antigen, not the other way…

Response: This has been described, and a review article has been cited as the DNA vaccine is still to be optimized as the outcome of animal models could not be translated into patients’ bedsides. Two references (References 51 and 52 have been cited to validate the claims of the investigators). However, this is an essential innovative therapy for CHB.

The studies about HBsAg-pulsed DC have also been explained in more detail in the revised manuscript.

Line 166: explain the limitations

This is partially due to the fact that there is a paucity of information about the mechanisms of action of the HBsAg-based vaccine in CHB patients. In addition, the HBsAg-based vaccine is supposed to induce HBsAg-specific immunity in CHB patients. However, HBcAg-specific immunity seems essential for the therapeutic effect of vaccine therapy in CHB patients.

Lines 173-175: problem in texts. Why has there been no consensus? Explain!

Response: Drug development includes a complex procedure of a combination of “Concepts,” “Evidence,” “Preclinical studies with the animal model,” “phase I/II/III clinical trial,” and providing data of short, middle, and long-term prognosis. Vaccine-based therapies did not follow these procedures, with some exceptions. Immune therapy with HBsAg-based vaccine was accomplished mainly as a pilot study, observational study, or small-scale phase study. After providing initial data at the end of treatment (EOT), in most cases, there was a publication about long-term follow-up. The proper design of immune therapy should be first authenticated by concept; this should be followed by animal and preclinical studies.

Line 179: extreme heterogeneity… Explain!

Response; This has been explained, and the logic has been cited. 

Line 201-202: The fact that the mechanism of action has been elucidated is a limitation???

Response: This has been explained in the revised manuscript. 

Line 210: optimistic data: give the information about that data! What were the results?

Response: The results have been provided in the revised manuscript.

Line 252-254. I do not agree that NUCs do not stop progression to CHC. Proofs must be given.

Response: This sentence has been modified on the basis of your suggestion.

Round 2

Reviewer 2 Report

Please see all my comments directly in the attached file below. Since there were no line numbers and page numbers I have commented within the PDF file.

Author Response

Response to Reviewer 2, report 2

Thank you very much for your constructive comments. A point to point response is given below.

Page 2:  “The WHO mentioned that in 2015, less than 1% of deserving CHB patients were under some treatment.”

Query of the Reviewer: This is from 2015. Either delete or include a recent study.

Response. This has been deleted as there is no world-wide data about this in literatures.

Page 2: Thus, the target indicates that, according to the WHO, an estimated 35.5–70 million CHB patients deserve treatment, and thus treatment should be expanded to an estimated 30-60 million CHB patients worldwide by 2030 (Figure 1B).

Query of the Reviewer: Rephrase

Response of the author: This has been accomplished according to the recommendation of the Reviewer.

Section: 4. The Advent of Vaccine Therapy for Treating CHB Patients

Query of the Reviewer: Immune Modulators: Give example for this.

Response of the Reviewer: Examples have been given.

Subsection .2. Different Forms of HBsAg-based Vaccine Therapy for Treating CHB Patients and their Limitations 

Query of the Reviewer: HBsAg was pulsed with antigen-presenting dendritic cells”- Probably, you wanted to delete this

Response of the Author: Thank you very much. This has been deleted.

Query of the Reviewer: Why successful? It is not clear why? Give examples. How many people eliminated HBV DNA upon treatment.

Response of the authors: The expression has been optimized by deleting “successful”

Query of the Reviewer: Spelling of parental

Response of the Authors: We are extremely sorry for the mis-selling. This has been corrected.

Query of the Reviewer: Anti-HBeAg

Response of the Authors: This has been corrected.

Query of the Reviewer: So, there was viral suppression? cccDNA?

Response of the Authors: This has been corrected.

Query of the Reviewer: The best proof of HBV elimination is cccDNA disappearance

Response of the Authors: We completely agree with the Reviewer. The expression has also been optimized.

Round 3

Reviewer 2 Report

The authors have considerably improved the manuscript. However, one modification should be made: figure 1: Remove from [B] "CHB patients under some treatment in 2005 <1% of chronic CHB patients"

Author Response

Response to Reviewer 2

Thanks very much for your constructive comment.

The suggestion of the Reviewer: The authors have considerably improved the manuscript. However, one modification should be made: figure 1: Remove from [B] "CHB patients under some treatment in 2005 <1% of chronic CHB patients".

Response of the Author: As per the suggestion of Reviewer 2, the phrase “CHB patients under some treatment in 2005 <1% of chronic CHB patients” in Figure 1. B has been deleted.
